# Architecturally Aligned Comparisons Between ConvNets And Vision Mambas

## Abstract

Mamba, an architecture with token mixers of state space models (SSM), has been recently introduced to vision tasks to tackle the quadratic complexity of self-attention. However, since SSM's memory is inherently lossy and precedent vision mambas struggle to compete with advanced ConvNets or ViTs, it is unclear whether Mamba has contributed new advances to vision. In this work, we carefully align the macro architecture to facilitate direct comparisons of token mixers which are the core contribution of Mamba. Specifically, we construct a series of Gated ConvNets (GConvNets) and compare VMamba's(Liu et al., 2024) token mixers with gated $7\times7$ depth-wise convolutions. The empirical results clearly demonstrate the superiority of VMamba's token mixers in both image classification and object detection tasks. Therefore, it is not useless to introduce SSM for image classification on ImageNet. Furthermore, we compare two types of token mixers within hybrid architectures that incorporate a few self-attention layers in the top blocks. The results demonstrate that both VMambas and GConvNets benefit from incorporating self-attention and we still need Mamba in this case. Interestingly, we find that incorporating self-attention layers has opposite effects on them, mitigating the over-fitting in VMambas while enhancing the fitting ability of GConvNets. Finally, we assess natural robustness of pure and hybrid models in image classification, revealing stronger robustness of VMambas and hybrid models. Our work provides credible evidence for the necessity of introducing Mamba to vision and shows the significance of architecturally aligned comparisons for evaluating different token mixers in sophisticated hierarchical models.

## 1 Introduction

For a considerable time, convolutional neural networks (CNNs)(LeCun et al., 1989; 1998) have been the primary neural networks in the vision domain. Notably, the success of AlexNet(Krizhevsky et al., 2012) in 2012 ushered in an era of deep learning in computer vision. Since then, various CNN architectures have been proposed, with representative networks such as VGG(Simonyan & Zisserman, 2014), GoogLeNet(Szegedy et al., 2015), ResNet(He et al., 2016), DenseNet(Huang et al., 2017; 2019), ResNeXt(Xie et al., 2017) and Xception(Chollet, 2017) having a significant impact on subsequent CNN architecture design. The success of convolutions can be attributed to their inherent inductive biases (locality and translation equivariance) and the sliding window strategy, which makes them robust to image resolution.

The dominance of CNNs in image recognition was not challenged until the introduction of Vision Transformers(Dosovitskiy et al., 2020). Inspired by the scalability of Transformers(Vaswani et al., 2017) in natural language processing (NLP), Dosovitskiy *et al.* apply a standard Transformer directly to images. Although ViTs lack some of the inductive biases inherent to CNNs, they attain excellent results when pre-trained on large-scale datasets such as ImageNet-21k, learning transferable features. Subsequent works improve the data efficiency(Touvron et al., 2021) and introduce image-related inductive biases, such as multi-scale(Wang et al., 2021; Fan et al., 2021; Liu et al., 2021; Wu et al., 2022) and locality(Liu et al., 2021; Wu et al., 2021; Yuan et al., 2021). These improved ViTs not only achieve state-of-the-art results on large-scale image recognition benchmarks but also significantly improve the performance of downstream tasks, such as detection and segmentation, compared to previous CNN based methods.

The success of ViTs draws researchers' attention to the underlying reasons for their effectiveness. Intuitively, this success is attributed to larger receptive fields and the dynamic feature modeling provided by self-attention mechanism. However, Yu et al. (2022) emphasize the importance of macro architecture, specifically the token mixer followed by the MLP. They show that the token mixer can be implemented as depthwise convolutions or even non-parametric average pooling. Meanwhile, ViTs face challenges from ConvNets with larger kernel sizes(Liu et al., 2022; Ding et al., 2022). The resurgence of ConvNets and the evolution of ViT architectures underscore the significance of inductive biases in convolutions.

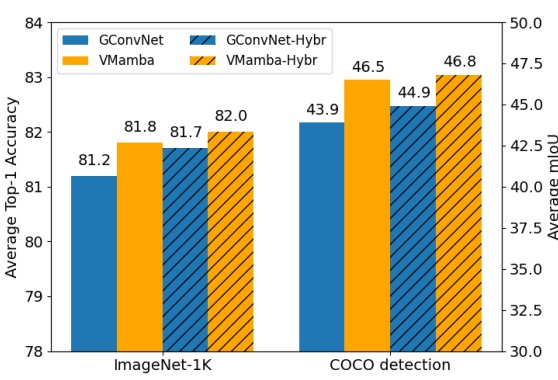

Figure 1: Results of architecturally aligned comparisons. Every result is the average result of models in three sizes.

Recently, Mamba(Gu & Dao, 2023), an RNN-like model, achieves highly competitive performance compared to Transformers in NLP while maintaining linear complexity relative to the number of tokens. Subsequently, several pioneering works migrate Mamba from language to vision, resulting in Vision Mamba models(Zhu et al., 2024; Liu et al., 2024; Li et al., 2024b; Huang et al., 2024). Nevertheless, the performance of Vision Mambas is often underwhelming compared to convolutional and attention-based models, prompting Yu & Wang (2024) to question whether we really need Mambas for vision. They conclude that Mambas are not needed for image classification, asserting "Mamba out". They argue that Mamba is ideally suited for tasks with long-sequence and autoregressive characteristics while image classification does not align with either characteristic. However, it remains puzzling why MambaOut outperforms VMamba(Liu et al., 2024) in image classification while significantly lagging behind in object detection and semantic segmentation. Importantly, we note that there are two architectural differences between the MambaOut models and the compared VMamba models, as illustrated in Fig. 2. Therefore, it is unclear whether the superiority of MambaOut models arises from their macro architecture or the gated 7×7 convolution. While contemporary Vision Mambas achieve superior accuracy or efficiency(Shi et al., 2024; Xiao et al., 2024; Hatamizadeh & Kautz, 2024), variations in architectural hyper-parameters, increasingly complex modules, and mixtures of self-attention layers leave the answer still unclear.

In light of the rapid increase in research in this area, we believe that an aligned comparison between Vision Mambas and their counterparts is urgently needed. Our focus is on hierarchical models, which have been shown to be more suitable for vision tasks than plain models. In this work, we conduct architecturally aligned comparisons between ConvNets and Vision Mambas, giving a credible answer to the question, "Do we really need Mamba for vision?" We select VMamba(Liu et al., 2024) as our reference model as it is one of the earliest works to adapt Mamba for the vision domain and serves as the main reference in MambaOut(Yu & Wang, 2024). To control architectural variables, we maintain the macro architecture of VMamba(Liu et al., 2024) while introducing GConvNet in different sizes, where the 2D Selective Scan (SS2D)(Liu et al., 2024) modules are replaced with gated 7×7 depth-wise convolutions. Our comparisons reveal a different conclusion than that of Yu & Wang (2024); our experimental results suggest that VMambas consistently outperform GConvNets on the ImageNet-lK benchmark with similar sizes or GFLOPs, as shown in Fig. 1. We hypothesize that this superiority is due to the stronger expressivity of VMamba's token mixers, which can be observed from training losses on ImageNet-1K. In object detection and instance segmentation tasks, VMambas significantly outperform GConvNets, highlighting the advantage of Mamba's token mixers in long-sequence modeling. To identify what makes MambaOut models superior to GConvNet and VMambas, we conduct further comparative experiments, showing that the MLP classifier is key to MambaOut's enhanced performance.

Furthermore, we demonstrate that incorporating a few self-attention layers in the top blocks improves the performance of both GConvNets and VMambas while the improvements on VMambas are relatively small, as shown in Fig. 1. Notably, VMamba-Hybrid clearly outperforms GConvNet-

Hybrid on COCO datasets, indicating that we still need Mamba in the presence of a few self-attention layers. Thanks to strictly aligned comparisons, we can take a deeper look. Specifically, we find that self-attention plays opposite roles in enhancing the performance of GConvNets and VMambas on ImageNet-1K: while adding self-attention layers enhances the fitting ability of GConvNets, it reduces over-fitting in VMambas. Finally, we compare GConvNet, VMamba, GConvNet-Hybrid, and VMamba-Hybrid in natural robustness of image classification, revealing stronger robustness of VMambas and hybrid models.

Our main contributions can be summarized as follows:

*(i)* We provide credible evidence for the necessity of introducing Mamba to vision, revealing the better performance of VMamba's token mixers on ImageNet-1K and COCO datasets, their stronger expressivity, and superior robustness compared to gated $7{\times}7$ depth-wise convolutions.

*(ii)* We show that incorporating a few self-attention layers cannot bridge the gap between ConvNets and Vision Mambas and the latter can also benefit from hybrid architectures. We further find that incorporating self-attention can mitigate the over-fitting in VMambas on ImageNet, providing evidence for the improved scalability of Vision Mamba-Transformer models.

*(iii)* We demonstrate the significance of architecturally aligned comparisons for evaluating different token mixers in sophisticated hierarchical models, a perspective often overlooked in previous research on model comparisons.

## 2 PRELIMINARIES

### 2.1 STATE SPACE MODELS

The mathematical foundations of Mambas' token mixers are state space models(Gu et al., 2021). The discrete forms of SSM can be expressed by:

$$
\begin{aligned}
h_t &= \overline{\mathbf{A}} h_{t-1} + \overline{\mathbf{B}} x_t, \\
y_t &= \mathbf{C} h_t, \\
\overline{\mathbf{A}} &= \exp(\Delta \mathbf{A}), \\
\overline{\mathbf{B}} &= (\Delta \mathbf{A})^{-1}(\exp(\Delta \mathbf{A}) - \mathbf{I}) \cdot \Delta \mathbf{B},
\end{aligned}
\tag{1}
$$

where $x_t$ represents the input, $h_t$ is the hidden state, $y_t$ indicates the output, and $\mathbf{A}, \mathbf{B}, \mathbf{C}$ are parameters of the continuous system. To improve the expression ability, Mamba(Gu & Dao, 2023) introduces the selective SSM where $\Delta, \mathbf{A}, \mathbf{B}, \mathbf{C}$ in Equation 1 are input-dependent parameters.

### 2.2 VISUAL STATE SPACE MODELS

The causal constraints of Mambas' token mixers render them unsuitable for processing images. To this end, Zhu et al. (2024) propose the bidirectional state space model and Liu et al. (2024) propose the 2D selective scan module which indeed comprises two bidirectional scanning: $H$-first scanning and $W$-first scanning. Subsequent works introduce the window-based local scanning strategy(Huang et al., 2024) and the continuous 2D scanning(Yang et al., 2024). In the context of this work, we consider VMamba(Liu et al., 2024) as a representative of Vision Mambas due to its prescience and influence.

## 3 METHOD

### 3.1 GCONVNET

The necessity of Mamba for vision should depend on the token mixer rather than other factors. Inspired by MambaOut(Yu & Wang, 2024), we investigate whether the token mixers in VMambas can be replaced by gated $7{\times}7$ depth-wise convolutions without degrading performance. A key distinction from Yu & Wang (2024) is our strict control over other architectural variables. Specifically, we replace the SS2D modules in VMamba(Liu et al., 2024) with gated $7{\times}7$ depth-wise convolutions, creating a fully convolutional network called GConvNet. The macro architectures of VMamba, our

GConvNet, and MambaOut are illustrated in Fig. 2. The model configurations for VMamba and GConvNet are detailed in Table 1, where we control for irrelevant variables such as the number of parameters, FLOPs, and depth-width trade-off. We compare six models in different sizes, from 8M to 50M parameters. Note that increasing network depth while reducing width typically yields better performance on ImageNet-1K, which we carefully control in our configurations.

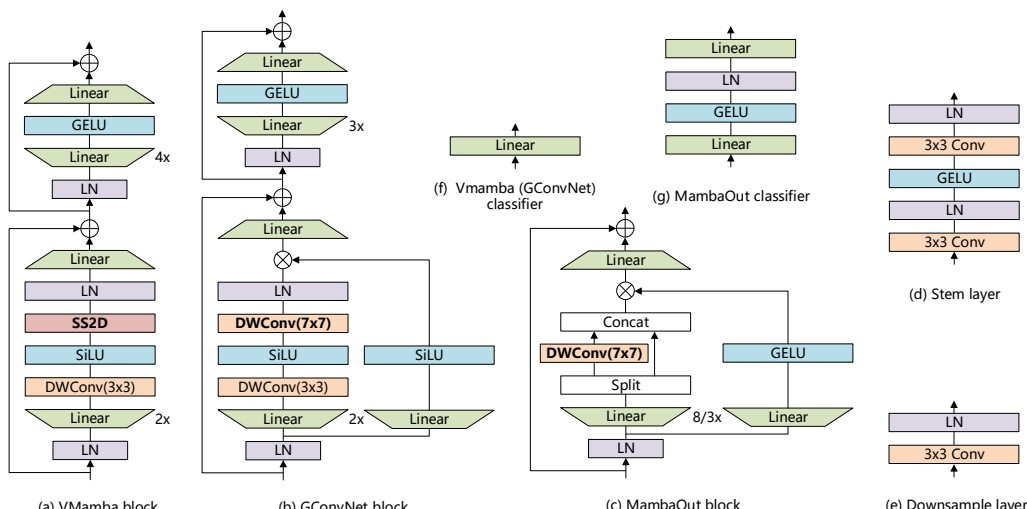

Figure 2: The macro architectures of VMamba, our GConvNet, and MambaOut are outlined with key variables highlighted in bold. To clarify how we control architectural variables, we divide the model architecture into four parts: the meta block (a)(b)(c), the stem layer (d), the downsample layer (e), and the classifier (f)(g). We present detailed structures of different meta blocks while omitting reshape operations. The VMamba block shown is from VMambaV9(Liu et al., 2024), consistent with that in MambaOut(Yu & Wang, 2024). There are two significant uncontrolled variables between VMamba and MambaOut: the structure of the meta block and the classifier. Note that in a MambaOut block, token mixers and channel mixers are arranged in parallel rather than sequentially. By contrast, the differences between VMamba and GConvNet are limited to the token mixers and the gated branch. While Liu et al. (2024) remove the gated branch as the SS2D module already provides dynamic modeling capabilities, we retain it in the GConvNet block. To control parameters and computation of point-wise linear layers, we reduce the expand ratio of FFN from 4.0 to 3.0.

Table 1: The model configurations of GConvNet and VMamba. Due to the alignment of meta blocks, we can adopt similar depth-width configurations to VMamba. Since the SS2D module has more parameters and computation than $7 \times 7$ depth-wise convolutions with the same width, we slightly increase the depths of GConvNet models to control the overall parameters and computation.

| Model | Layers | Dims | Params | GFLOPs |
|---|---|---|---|---|
| VMamba-Pico | [2, 2, 5, 2] | [48, 96, 192, 384] | 7.9M | 1.27G |
| VMamba-Tiny | [2, 2, 5, 2] | [96, 192, 384, 768] | 30.7M | 4.86G |
| VMamba-Small | [2, 2, 15, 2] | [96, 192, 384, 768] | 50.1M | 8.72G |
| GConvNet-Pico | [2, 2, 6, 2] | [48, 96, 192, 384] | 8.0M | 1.27G |
| GConvNet-Tiny | [2, 2, 6, 2] | [96, 192, 384, 768] | 30.8M | 4.88G |
| GConvNet-Small | [2, 2, 17, 2] | [96, 192, 384, 768] | 50.8M | 8.79G |

## 3.2 HYBRID MODELS WITH A FEW TRANSFORMER BLOCKS

Previous works have shown that performing convolutions in the bottom blocks to extract local information while applying self-attention layers in the top blocks to model global relationships, can

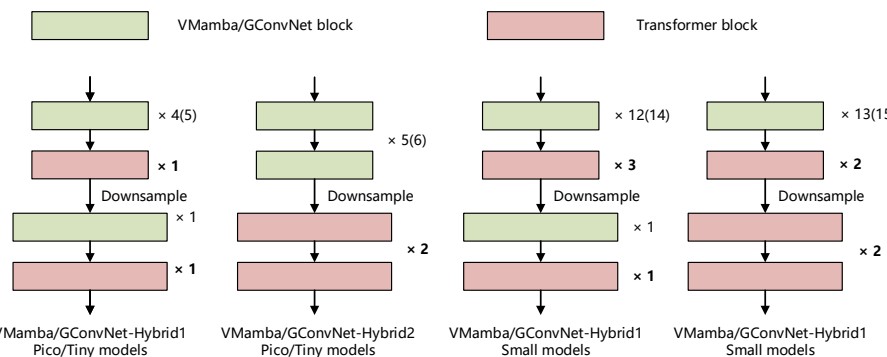

Figure 3: Two kinds of mixing strategies. "Hybrid1" ensures that there is at least one self-attention layer at resolution 1/16 while "Hybrid2" is more economically.

yield superior performance(Dai et al., 2021; Yu et al., 2023). Recently, Dao & Gu (2024) demonstrate that a mixture of Mamba-2 token mixers and attention layers outperforms the pure Mamba-2 or Transformer architecture, indicating the complex principles behind hybrid models. This inspires us to investigate the effect of integrating a few self-attention layers with GConvNets and VMambas and compare these hybrid models. We emphasize the limited number of self-attention layers because our goal is to compare convolutions and SSM which are two economical substitutes for self-attention in vision. We follow Dao & Gu (2024) to replace approximately 10-20% GConvNet or VMamba blocks with Transformer blocks. Specifically, pico models and tiny models include 2 Transformer blocks while small models incorporate 4 Transformer blocks. We examine two mixing strategies to understand the principles of this integration. The first involves replacing the top VMamba or GConvNet blocks in the last two stages proportionally, while the second replaces blocks from top to bottom. The former generally results in more self-attention layers at resolution 1/16 compared to the latter. We illustrate these two strategies in Fig. 3. The vanilla Transformer block with CPE(Chu et al., 2023) is employed, which can be expressed as:

$$
\begin{aligned}
x &= \text{DWConv}_{3\times3}(x) + x \\
x &= \text{MSA}(\text{LayerNorm}(x)) + x, \\
x &= \text{FFN}(\text{LayerNorm}(x)) + x,
\end{aligned}
\tag{2}
$$

where MSA denotes the multi-head self-attention and FFN represents the feed forward network made up of two linear layers and a GELU activation. The expand ratio of FFNs is set to 4.

## 4 EXPERIMENTAL SETUPS

We primarily conduct experiments on ImageNet-1K(Deng et al., 2009) and COCO(Lin et al., 2014) datasets. The former is used to evaluate the performance in image classification tasks while the latter assesses transferability in object detection and instance segmentation tasks. Both are widely recognized benchmarks. For ImageNet-1K, we adopt the same training and test protocols as VMamba, with the sole difference being the absence of EMA(Polyak & Juditsky, 1992), which does not improve performance. Thus, our protocols align with those of Swin(Liu et al., 2021). For COCO, we use the same codebase based on MMdetection(Chen et al., 2019) and directly replace backbone networks. For robustness evaluation in image classification, we follow previous works(Zhou et al., 2022; Bhojanapalli et al., 2021) and assess models across three datasets: ImageNet-A(Hendrycks et al., 2021b), ImageNet-R(Hendrycks et al., 2021a), and ImageNet-C(Hendrycks & Dietterich, 2019). Detailed experimental setups are provided in the Appendix.

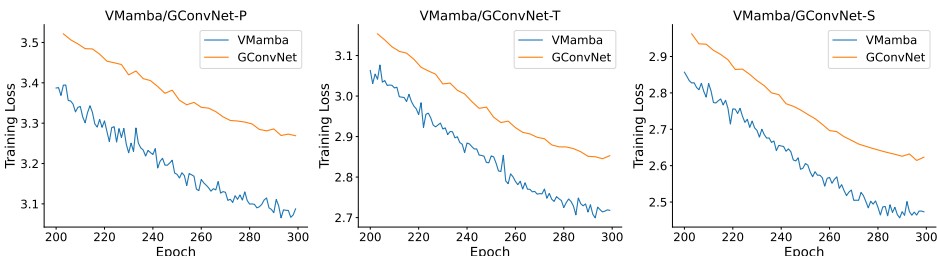

Figure 4: Training loss of VMamba and GConvNet. For higher efficiency, we evaluate GConvNet every three epochs during training.

## 5 RESULTS AND ANALYSES

### 5.1 DO WE REALLY NEED MAMBAS FOR VISION?

**It is not useless to introduce SSM for image classification on ImageNet.** As shown in Fig and Table 2, VMamba clearly outperforms GConvNet on both ImageNet-1K and COCO datasets. This suggests that in image classification tasks, the well-designed SSM can be superior to gated $7\times7$ depth-wise convolutions which advance ConvNets for the 2020s. The advantage is even more pronounced in smaller models. Consequently, we challenge a critical hypothesis of MambaOut(Yu & Wang, 2024): it is not useless to introduce SSM for image classification on ImageNet. These results provide credible evidence supporting the recent advancements in Mambas for vision. We hypothesize that this superiority is due to the stronger expressivity of Mambas' token mixers. It can be observed from training loss curves in Fig. 4 where VMambas exhibit lower training losses on ImageNet compared to GConvNets.

Table 2: Performance comparisons between GConvNets and VMambas on ImageNet-1K and COCO. The results of VMambas are obtained by the best checkpoints rather than the last checkpoints following the original paper(Liu et al., 2024). We present the results of the last checkpoints in parentheses. ∗: our reproduced result is slightly better than the result (82.5) reported by Liu et al. (2024).

| Model | Top-1 accuracy | $AP^b$ | $AP^m$ |
|---|---|---|---|
| VMamba-Pico | **79.1** (79.0) | **43.4** | **39.7** |
| GConvNet-Pico | 78.4 | 40.8 | 37.5 |
| VMamba-Tiny | **82.6** (82.5)∗ | **47.1** | **42.6** |
| GConvNet-Tiny | 82.2 | 44.7 | 40.5 |
| VMamba-Small | **83.6** (83.1) | **49.0** | **43.7** |
| GConvNet-Small | 83.1 | 46.1 | 41.5 |

**Vision Mambas have more potential in lightweight object detection models.** Lightweight models usually suffer from limited expressivity and receptive fields, which are crucial for more difficult downstream tasks including detection and segmentation. The strong expressivity and truly global receptive fields of Vision Mambas probably make them excel in lightweight object detection. In Table 3, we show that without tuning depth-width configurations or specific designs, VMamba-Pico with fewer parameters can compete with state-of-the-art lightweight models that combine convolutions and self-attention. The best-performance EfficientMod-s(Ma et al., 2024) utilizes 4 vanilla transformer blocks at resolution 1/16 and 4 vanilla transformer blocks at resolution 1/32, which will suffer from the quadratic complexity of self-attention when the input resolution is very large.

Table 3: Performance of lightweight backbones on COCO.

| Arch. | Backbone | Params | $AP^b$ | $AP^m$ |
|---|---|---|---|---|
| Conv. | ResNet-18 (2016) | 31.2M | 34.0 | 31.2 |
| Pool | PoolF.-S12 (2022) | 31.6M | 37.3 | 34.6 |
| Attn. | PVT-Tiny (2021) | 32.9M | 36.7 | 35.1 |
| Conv-attn. | EfficientF.-L1 (2022) | 31.5M | 37.9 | 35.4 |
| Conv-attn. | PVTv2-B1 (2022) | 33.7M | 41.8 | 38.8 |
| Conv-attn. | EfficientF.V2-S2 (2023) | 32.2M | 43.4 | 39.5 |
| Conv-attn. | EfficientMod-s (2024) | 32.6M | **43.6** | **40.3** |
| Mamba | VMamba-P | **27.6M** | 43.4 | 39.7 |

## 5.2 What Makes MambaOut Excel in Image Classification?

**The MLP classifier is key to the superior performance of MambaOut on ImageNet.** We have disassembled the network architecture in Fig. 2. We then exclude the MLP classifier and use the MambaOut block (or Gated CNN block) to construct local MambaOut models. Note that once the MLP classifier is replaced by the linear classifier, we adjust the dimension of the last stage to a conventional value of 768, instead of the original 576 in MambaOut-Tiny. This change results in more model parameters and computation. The results of our local MambaOut model are shown in the second line from the bottom of Table 4. It can be seen that the MLP classifier, rather than the block structure, is crucial for the superior performance of MambaOut on ImageNet-1K. The comparison between GConvNet-Tiny and MambaOut-Tiny without the MLP classifier suggests that our GConvNet block is not an inferior structure. At last, we apply the MLP classifier to VMamba and reduce the dimension of the last stage similarly to MambaOut, which also leads to improved performance and reduced computation. Since the MLP classifier essentially increases non-linearity and improves expressivity, the performance gain on VMamba is not as pronounced as that on MambaOut.

Table 4: An ablation of the macro architecture of MambaOut. ∗: we can reproduce the result of MambaOut-Tiny using our environments.

| Model | Params | GFLOPs | Top-1 accuracy | $AP^b$ | $AP^m$ |
|---|---|---|---|---|---|
| VMamba-Tiny | 30.7M | 4.86G | 82.6 | 47.1 | 42.6 |
| GConvNet-Tiny | 30.8M | 4.88G | 82.2 | 44.7 | 40.5 |
| MambaOut-Tiny | 26.5M | 4.47G | 82.7∗ | 44.6 | 40.4 |
| MambaOut-Tiny w/o MLP classifier | 30.6M | 4.81G | 82.1 | 44.9 | 40.8 |
| VMamba-Tiny w/ MLP classifier | 26.2M | 4.50G | **82.9** | 47.3 | 42.8 |

## 5.3 Do We Need Mambas in The Presence of A Few Self-attention Layers?

**Incorporating a few self-attention layers in the top blocks improves the performance of both GConvNets and VMambas. Introducing SSM remains beneficial even in the presence of a few self-attention layers, particularly for downstream long-sequence tasks.** We first examine two mixing strategies in Fig. 3 using pico and tiny models. From Table 5, we observe that incorporating self-attention layers in GConvNet consistently improves performance on ImageNet-1K and COCO datasets. Additionally, GConvNet-Hybrid1 outperforms GConvNet-Hybrid2 overall, suggesting that applying self-attention at a higher resolution yields greater benefits, akin to the findings in BotNet(Srinivas et al., 2021). Nonetheless, our research focuses on more advanced ConvNets with larger kernel sizes and gated mechanisms rather than vanilla ResNets. In contrast, both mixing strategies yield minimal gains for VMamba-Pico and VMamba-Tiny on ImageNet-1K, with slight improvements on COCO. For subsequent fair comparisons, we adopt the first mixing strategy by default and train larger models. The performance of GConvNet-Hybrid-Small meets expectations while VMamba-Hybrid-Small shows significant improvement on ImageNet-1K. Although GConvNets-

Hybrid can achieve performance comparable to VMambas on ImageNet-1K, they still lag behind in object detection and instance segmentation tasks. Comparing GConvNet-Hybrid and VMamba-Hybrid, we believe it is still useful to introduce SSM in the presence of a few self-attention layers, especially for downstream long-sequence tasks.

Table 5: Performance of hybrid models on ImageNet-1K and COCO. We show how the performance of hybrid models varies compared to pure counterparts in the parentheses.

| Model | Top-1 accuracy | $AP^b$ | $AP^m$ |
|---|---|---|---|
| VMamba-Pico | 79.1 | 43.4 | 39.7 |
| GConvNet-Hybrid1-Pico | 78.9 (+0.5) | 41.6 (+0.8) | 38.3 (+0.8) |
| GConvNet-Hybrid2-Pico | 78.4 (+0.0) | 41.3 (+0.5) | 38.2 (+0.7) |
| VMamba-Hybrid1-Pico | 79.1 (+0.1) | 43.6 (+0.2) | 39.8 (+0.1) |
| VMamba-Hybrid2-Pico | 79.0 (-0.1) | 43.6 (+0.2) | 39.9 (+0.2) |
| VMamba-Tiny | 82.6 | 47.1 | 42.6 |
| GConvNet-Hybrid1-Tiny | 82.8 (+0.6) | 45.9 (+1.2) | 41.7 (+1.2) |
| GConvNet-Hybrid2-Tiny | 82.9 (+0.7) | 45.6 (+0.9) | 41.3 (+0.8) |
| VMamba-Hybrid1-Tiny | 82.6 (+0.0) | 47.7 (+0.6) | 43.0 (+0.4) |
| VMamba-Hybrid2-Tiny | 82.7 (+0.1) | 47.3 (+0.2) | 42.8 (+0.2) |
| VMamba-Small | 83.6 | 49.0 | 43.7 |
| GConvNet-Hybrid1-Small | 83.5 (+0.4) | 47.3 (+1.2) | 42.5 (+1.0) |
| VMamba-Hybrid1-Small | **84.2** (+0.5) | **49.1** (+0.1) | **43.8** (+0.1) |

**Incorporating self-attention layers in the top blocks reduces the over-fitting in VMambas while enhancing the fitting ability of GConvNets.** The unexpected gain of VMamba-Hybrid-Small prompts us to investigate the reason behind the superiority of SSM-attention hybrid models on ImageNet-1K. Our intriguing finding reveals that the advantages of GConvNet-Hybrid and VMamba-Hybrid compared to their pure counterparts stem from opposite effects. Specifically, adding self-attention layers in the top blocks reduces over-fitting in VMambas while enhancing the fitting ability of GConvNets. We present the training losses of VMamba, VMamba-Hybrid, GConvNet, and GConvNet-Hybrid on ImageNet-1K in Fig. 5. It can be seen that VMambas-Hybrid exhibit higher training losses than VMambas while GConvNets-Hybrid achieve lower train losses compared to GConvNets. Furthermore, we plot the curves of Top-1 (EMA) accuracy on ImageNet-1K against epochs for VMamba and VMamba-Hybrid in Fig. 6. The EMA accuracy curve of VMamba-Tiny hints at slight over-fitting as the performance peaks at epoch 242 and then slowly declines. This issue is more pronounced for VMamba-Small. Comparing the EMA accuracy curves of VMamba and VMamba-Hybrid also confirms that the over-fitting issues are mitigated. Importantly, the use of EMA itself can help reduce over-fitting in large models. Notably, without EMA, VMamba-Hybrid-Small surpasses VMamba-Small by 0.9 % in Top-1 accuracy. The over-fitting problems of Vision Mambas are also suggested by previous works(Zhu et al., 2024; Liu et al., 2024; Li et al., 2024a) where larger models may achieve inferior performance compared to smaller models. We clearly demonstrate that incorporating self-attention layers presents a promising architectural strategy for improving the scalability of Vision Mambas. Our finding also provides practical insights into when and how to incorporate self-attention layers effectively on ImageNet:

- For well-designed lightweight Vision Mamba models in under-fitting, it is unnecessary to incorporate self-attention layers.
- Self-attention layers should be added in the top blocks and incorporating more self-attention layers may not bring more performance gain, which involves a balance of fitting and generalization.

## 5.4 DO WE NEED MAMBA IN ROBUSTNESS?

**VMambas are generally more robust than GConvNets and incorporating self-attention layers typically enhances robustness.** In this section, we evaluate model robustness in image classification

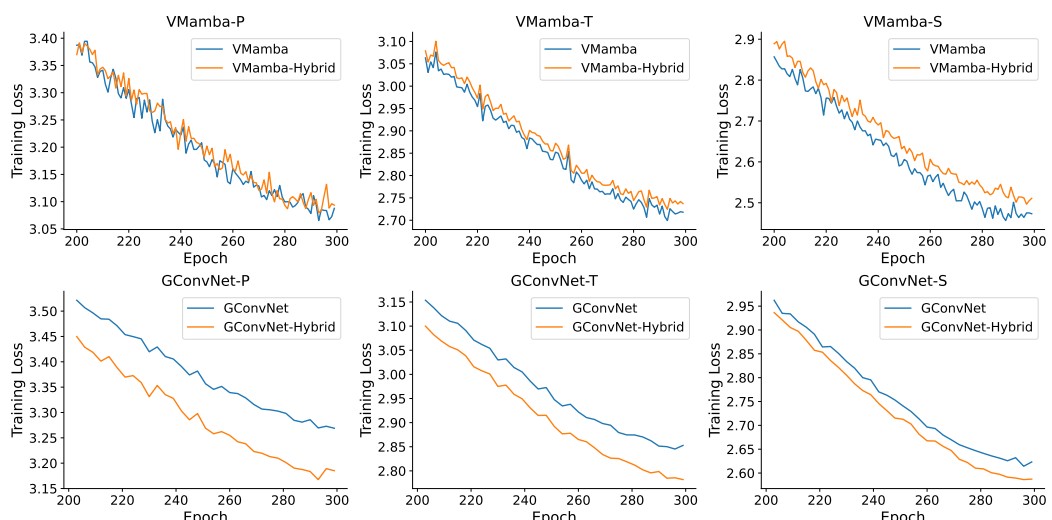

Figure 5: Training losses on ImageNet-1K vs epochs.

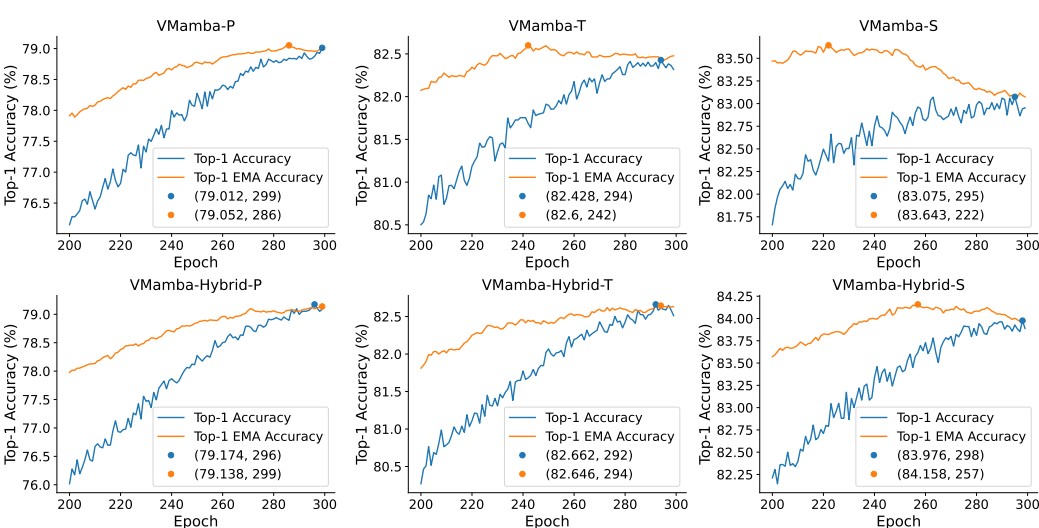

Figure 6: Top-1 (EMA) accuracy on ImageNet-1K vs epochs.

using three benchmarks. We focus on natural robustness, specifically, robustness to real-world images that can deceive pre-trained classifiers (indicated by Top-1 accuracy on ImageNet-A), robustness to various artistic renditions (indicated by Top-1 accuracy on ImageNet-R), and robustness to natural corruptions (indicated by mCE on ImageNet-C). We leave adversarial robustness for future work. Note that our goal is not to achieve leading results but to provide insights through aligned comparisons. All the results are presented in Fig. 7, which includes 12 contrasts. More detailed results are in the Appendix. From Fig. 7, we draw two key observations. Firstly, VMambas generally demonstrate greater robustness than GConvNets except for GConvNet-Tiny on ImageNet-A. Similarly, VMambas-Hybrid are more robust than GConvNets-Hybrid with the same exception for GConvNet-Tiny on both ImageNet-A and ImageNet-R. Notably, VMambas and VMambas-Hybrid consistently achieve lower mCE than their GConvNet counterparts on ImageNet-C, indicating stronger robustness of Vision Mambas to natural corruptions. Secondly, hybrid models typically exhibit greater robustness than their pure counterparts with the sole exception being VMamba-Hybrid-Tiny on ImageNet-R. Overall, incorporating self-attention layers improves the robustness of both VMambas and GConvNets.

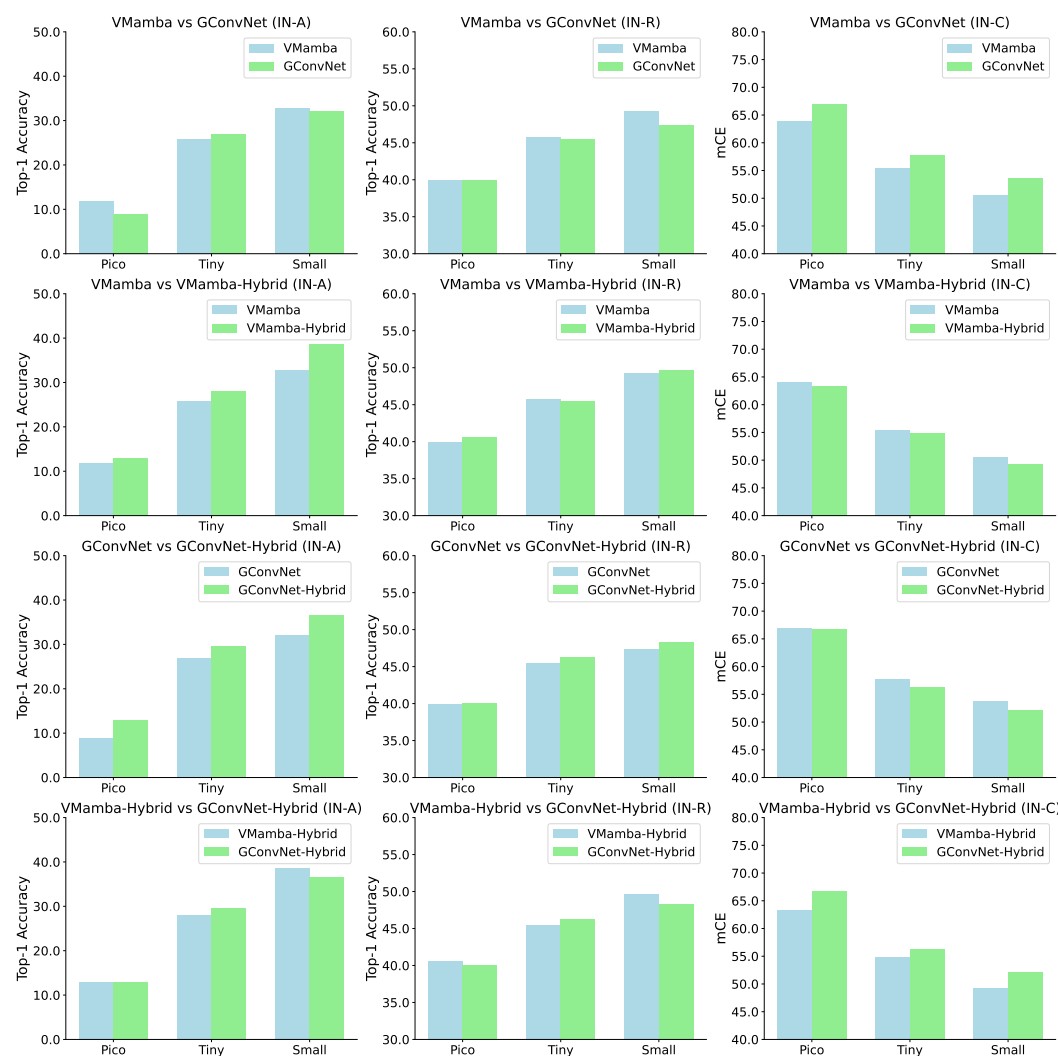

Figure 7: Robustness comparisons on ImageNet-A (IN-A), ImageNet-R (IN-R), and ImageNet-C (IN-C). Note that for mCE(Hendrycks & Dietterich, 2019), the lower is better. For fair comparisons, all the hybrid models adopt the first mixing strategy.

## 6   CONCLUSION

In this work, we conduct architecturally aligned comparisons between ConvNets and Vision Mambas, providing credible evidence for the necessity of introducing Mamba to vision. We reveal the better performance of VMamba's token mixers on ImageNet and COCO datasets, their stronger expressivity, and superior robustness compared to gated 7×7 depth-wise convolutions. We also show that incorporating a few self-attention layers cannot bridge the gap between ConvNets and Vision Mambas and the latter can also benefit from hybrid architectures. Additionally, we find that incorporating a few self-attention layers in the top blocks can mitigate over-fitting in VMambas on ImageNet, presenting a promising architectural strategy for improving the scalability of Vision Mambas. Considering that more token mixers from other fields such as NLP may be introduced into vision in the future, our work emphasizes the importance of aligned comparisons when combining them with sophisticated hierarchical models.

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

# A APPENDIX

## A.1 RELATED WORKS

Transformers have become standard components of high-performance vision backbones(Dosovitskiy et al., 2020; Fan et al., 2021; Liu et al., 2021; He et al., 2022; Shi, 2024). However, the quadratical complexity of self-attention layers makes vanilla ViTs struggle with high-resolution image processing. Consequently, many works propose various efficient self-attention mechanism by incorporating the inherent inductive biases of convolutions or images(Wang et al., 2021; Liu et al., 2021; Wu et al., 2022; Shi, 2024). Meanwhile, ConvNets for the 2020s emerge, sharing the block structure of Transformers while utilizing depth-wise convolutions with larger kernel sizes(Liu et al., 2022; Ding et al., 2022; Liu et al., 2023), achieving highly competitive performance compared to state-of-the-art ViTs.

To address the computational challenge of Transformers in processing long sequences, numerous works in the NLP field have explored various approaches, including RNN-like methods(Katharopoulos et al., 2020; Peng et al., 2023; Gu & Dao, 2023). Consequently, in addition to designing vision-specific efficient self-attention mechanisms, transferring these efficient token mixers with global modeling capacity to vision is also a promising direction. Recently, researchers have quickly introduced Vision Mambas(Zhu et al., 2024; Liu et al., 2024; Li et al., 2024b; Huang et al., 2024; Shi et al., 2024; Hatamizadeh & Kautz, 2024; Xiao et al., 2024), which incorporate SSM and Mambas(Gu & Dao, 2023) into vision backbones. Unlike previous works on Vision Mambas that focus on proposing novel modules, Yu & Wang (2024) present MambaOut models made up of simpler gated CNN blocks, comprehensively outperforming VMambas(Liu et al., 2024) on ImageNet-1K. However, there may be unfair comparisons that lead to an underestimation of Vision Mambas. In this work, we conduct aligned comparisons between ConvNets and Vision Mambas for the first time, provides credible evidence for the necessity of introducing Mamba to vision.

## A.2 EXPERIMENTAL SETUPS

**ImageNet-1K** For VMamba-Hybrid, the training protocols are identical to those of VMamba(Liu et al., 2024). For GConvNet and GConvNet-Hybrid, we remove the EMA(Polyak & Juditsky, 1992) as it does not improve the performance. All the models are trained from scratch for 300 epochs, with a warm up of 20 epochs, using a batch size of 1024. We utilize the AdamW optimizer with a momentum of 0.9, an initial learning rate of 0.001, and a weight decay of 0.05. The cosine scheduler is utilized to decay the learning rate. The drop path rate of pico, tiny, and small models are 0.025, 0.2, and 0.03.

**COCO** We follow VMamba(Liu et al., 2024) and Swin(Liu et al., 2021) to utilize the well-established Mask R-CNN framework(He et al., 2017) for evaluating the performance of object detection and instance segmentation. We also utilize the MMdetection(Chen et al., 2019) toolbox and all the hyper-parameters are identical to those of VMamba. Specifically, we employ the AdamW optimizer with an initial learning rate of 0.0001, load pre-trained weights of ImageNet-1K, and fine-tune the models for 12 epochs. Automatic Mixed Precision (AMP) is employed to accelerate training. The drop path rate of pico, tiny, and small models are 0.025, 0.2, and 0.03.

**ImageNet-C** This dataset(Hendrycks & Dietterich, 2019) totally contains 19 corrupted ImageNet-1K val sets. We evaluate the performance of models pre-trained on ImageNet-1K to benchmark robustness to natural corruptions. We primarily report mCE(Hendrycks & Dietterich, 2019) following previous works. The detailed Top-1 accuracy is shown in Section A.3. More details about the calculation of mCE can be found in its original paper.

**ImageNet-A** This dataset(Hendrycks et al., 2021b) is made up of real-world adversarially filtered images that can fool pre-trained classifiers on ImageNet. We evaluate the performance of models pre-trained on ImageNet-1K and report Top-1 accuracy following previous works.

**ImageNet-R** This dataset(Hendrycks et al., 2021a) comprises various artistic renditions of 200 classes from ImageNet-1K. We evaluate the performance of models pre-trained on ImageNet-1K and report Top-1 accuracy following previous works.

## A.3 DETAILED RESULTS ABOUT ROBUSTNESS

We present numerical results of robustness evaluation in Table 6 and detailed results on ImageNet-C in Table 7.

Table 6: Performance on ImageNet-A, ImageNet-R, and ImageNet-C.

| Model | IN | IN-A | IN-R | IN-C ↓ |
|---|---|---|---|---|
| GConvNet-Pico | 78.4 | 8.9 | 39.9 | 66.9 |
| GConvNet-Tiny | 82.2 | 27.0 | 45.5 | 57.8 |
| GConvNet-Small | 83.1 | 32.2 | 47.4 | 53.7 |
| VMamba-Pico | 79.1 | 11.8 | 40.0 | 64.0 |
| VMamba-Tiny | 82.6 | 25.7 | 45.8 | 55.5 |
| VMamba-Small | 83.6 | 32.8 | 49.3 | 50.6 |
| GConvNet-Hybrid-Pico | 78.9 | 12.9 | 40.1 | 66.7 |
| GConvNet-Hybrid-Tiny | 82.8 | 29.7 | 46.3 | 56.3 |
| GConvNet-Hybrid-Small | 83.5 | 36.6 | 48.3 | 52.1 |
| VMamba-Hybrid-Pico | 79.1 | 13.0 | 40.6 | 63.3 |
| VMamba-Hybrid-Tiny | 82.6 | 28.1 | 45.5 | 54.9 |
| VMamba-Hybrid-Small | **84.2** | **38.7** | **49.7** | **49.3** |

Table 7: Detailed results on ImageNet-C. "Aver" is the average Top-1 accuracy under 19 abnormal conditions.

| Model | Aver | Motion blur | Defoc blur | Glass blur | Gauss blur | Gauss noise | Impul noise | Shot noise | Speck noise | Contr | Satur | JPEG | Pixel | Bright | Snow | Fog | Frost | Zoom blur | Elastic trans | Spatter |
|---|---|---|---|---|---|---|---|---|---|---|---|---|---|---|---|---|---|---|---|---|
| | | | | | | | | | | | | GConvNet | | | | | | | | |
| Pico | 49.0 | 45.7 | 38.8 | 27.4 | 42.4 | 46.6 | 44.7 | 45.1 | 50.9 | 67.5 | 58.4 | 49.0 | 69.7 | 43.3 | 53.2 | 50.2 | 48.7 | 36.1 | 44.6 | 58.8 |
| Tiny | 56.1 | 52.6 | 45.5 | 31.5 | 48.0 | 56.3 | 56.5 | 54.7 | 60.2 | 63.6 | 72.6 | 63.7 | 58.3 | 74.5 | 50.9 | 58.1 | 57.6 | 45.6 | 50.2 | 64.8 |
| Small | 59.2 | 56.7 | 48.6 | 34.3 | 50.6 | 61.5 | 61.0 | 59.6 | 63.7 | 67.3 | 74.3 | 65.8 | 58.1 | 75.8 | 53.3 | 63.6 | 60.8 | 49.0 | 54.1 | 67.0 |
| | | | | | | | | | | | | VMamba | | | | | | | | |
| Pico | 51.3 | 46.8 | 42.4 | 27.0 | 45.1 | 50.6 | 48.4 | 48.6 | 54.0 | 58.8 | 69.3 | 60.6 | 51.9 | 71.3 | 45.3 | 55.5 | 51.9 | 38.6 | 47.2 | 60.3 |
| Tiny | 58.0 | 52.4 | 47.8 | 33.2 | 50.5 | 59.3 | 58.6 | 50.0 | 63.4 | 65.4 | 73.9 | 66.2 | 56.3 | 75.6 | 53.5 | 62.7 | 59.4 | 45.3 | 53.2 | 66.4 |
| Small | 61.6 | 58.4 | 52.5 | 37.1 | 54.8 | 62.8 | 62.1 | 61.3 | 66.1 | 68.3 | 75.4 | 67.8 | 61.8 | 76.9 | 57.7 | 67.4 | 61.7 | 51.8 | 57.4 | 69.4 |
| | | | | | | | | | | | | GConvNet-Hybrid | | | | | | | | |
| Pico | 49.2 | 46.0 | 39.7 | 27.0 | 43.0 | 46.6 | 45.2 | 44.2 | 50.9 | 56.7 | 68.6 | 59.4 | 45.9 | 70.7 | 44.6 | 55.2 | 51.2 | 36.4 | 45.2 | 59.0 |
| Tiny | 57.3 | 52.8 | 46.7 | 31.5 | 49.1 | 58.3 | 58.2 | 56.7 | 62.1 | 64.3 | 73.5 | 64.6 | 57.2 | 75.4 | 52.7 | 62.8 | 59.0 | 45.4 | 51.9 | 67.2 |
| Small | 60.4 | 58.1 | 50.0 | 34.1 | 52.1 | 61.8 | 62.6 | 59.6 | 63.9 | 67.6 | 74.9 | 66.8 | 60.1 | 76.7 | 55.4 | 68.0 | 62.5 | 50.0 | 54.2 | 68.6 |
| | | | | | | | | | | | | VMamba-Hybrid | | | | | | | | |
| Pico | 51.8 | 45.4 | 42.9 | 28.4 | 45.9 | 50.8 | 50.0 | 48.8 | 54.2 | 60.6 | 69.3 | 60.5 | 52.7 | 71.5 | 47.7 | 55.9 | 52.7 | 37.8 | 48.2 | 60.9 |
| Tiny | 58.4 | 53.6 | 48.8 | 33.3 | 51.4 | 58.8 | 58.9 | 57.1 | 62.3 | 65.1 | 74.1 | 66.0 | 58.4 | 75.6 | 55.1 | 64.4 | 59.9 | 46.7 | 53.0 | 67.1 |
| Small | 62.5 | 60.6 | 53.1 | 38.1 | 55.3 | 64.4 | 64.4 | 62.1 | 66.0 | 67.9 | 75.9 | 68.7 | 64.2 | 77.3 | 57.9 | 68.8 | 62.6 | 53.3 | 57.4 | 69.8 |

