# OpenReview forum: "Architecturally Aligned Comparisons Between ConvNets And Vision Mambas"
_ICLR.cc/2025/Conference — Submitted to ICLR 2025_

### Official Review · Reviewer_ty4j · 2024-10-16

**Soundness:** 3
**Presentation:** 3
**Contribution:** 2
**Rating:** 5
**Confidence:** 4

**Summary:**

This paper aligned the macro architecture to facilitate direct comparisons of token mixers which are the core contribution of Mamba.

**Strengths:**

1. This paper is well-written and easy to follow.
2. The experimental results verify the effectiveness of the proposed method to some extent.

**Weaknesses:**

1. The motivation is unclear. First, it is unclear what visual task-oriented methods are based on Mamba. Second, it is unclear what problems exist in the above methods. Finally, the differences and advantages between the technique in this paper and the above methods should be further supplemented.
2. There is a lack of classification and summary of existing related work.
3. The proposed method lacks rationality. The proposed GConvNet method is like a stack of existing network modules. It is unclear why these modules are stacked like this.
4. Experimental organization and analysis are missing. First, it is unclear why the experiments are organized in questions 5.1 to 5.4. What is the relationship between these questions? Second, the analysis is insufficient for ablation studies. It is unclear why each of the proposed components can bring performance improvements.

**Questions:**

See the comments below.

---

### Official Review · Reviewer_P7hm · 2024-10-31

**Soundness:** 3
**Presentation:** 3
**Contribution:** 2
**Rating:** 5
**Confidence:** 4

**Summary:**

To explore whether mamba has contributed new advances to computer vision, this paper aligns the macro architectures of ConvNets and vision mambas to facilitate direct comparisons between them. Specifically, this paper constructs a series of gated ConvNets and compares VMamba with gated 7x7 depth-wise convolutions. Furthermore, ConvNets and vision mambas are compared within hybrid architectures that incorporate a few self-attention layers in the top blocks. Finally, the natural robustness of pure and hybrid models is assessed in image classification.

**Strengths:**

This paper compares vision mambas with ConvNets in several ways, which may be useful for future research in vision mambas. The paper is well-motivated. The writing is professional and clear. Additionally, the paper is well-organized, with clear and informative figures and tables that enhance the reader’s understanding of the data and key trends.

**Weaknesses:**

This paper only provides comparisons between vision mambas and ConvNets. However, vision transformers have achieved state-of-the-art performance in various vision tasks and thus need to be compared and discussed. It is also interesting to discuss and evaluate vision transformers + convolutions and vision transformer + vision mambas, where vision transformers are the main blocks, rather than vision mambas / convolutions as shown in Figure 3.

The convolution-based model, GConvNet, is designed by simply modifying the VMamba block. GConvNet performs a little bit worse than VMamba. However, convolution-based models and vision mambas may have different optimal architectures. The poor performance of GConvNet may be not because convolution is a worse token mixer than vision mamba, but because the convolution-based model, GConvNet, is not in its optimal case. Therefore, I do not agree with the architecturally aligned comparisons used in this paper.

This paper only evaluates one variant of vision mambas, VMamba, and one self-developed convolutional network, GConvNet. This is very limited for demonstrating the general characteristics of vision mambas and convolutional networks. To convince the readers, more mamba models and enough successful convolutional networks should be evaluated, compared, and discussed.

This paper only evaluates some small networks. I would like to see the results of larger models because large models may have different characteristics from small models.

This paper evaluates the characteristics of vision mambas and convolutional networks. With such experiments and discussions, I would expect the paper to develop a new model that achieves state-of-the-art performance. This is essential to demonstrate the usefulness of the findings. However, there is no new model and even no new techniques found in this paper. Thus, the paper lacks technical novelty.

The writing should be checked carefully. Here are some suggestions: abbreviations need to be defined before use (like ViTs). There should not be references in the abstract (like “VMamba’s(Liu et al., 2024)”). There should be a space between the text and its following references (like “AlexNet (Krizhevsky et al., 2012)”, not “AlexNet(Krizhevsky et al., 2012)”). An example is as follows:

“However, since SSM’s memory is inherently lossy and precedent vision mambas struggle to compete with advanced ConvNets or ViTs, it is unclear whether Mamba has … to vision.” -> “However, since SSM’s memory is inherently lossy and precedent, vision mambas struggle to compete with advanced ConvNets or vision transformers (ViTs), it is unclear whether mamba has … to computer vision.” (“precedent” -> “precedent,”, “ViTs” -> “vision transformers (ViTs)”, “Mamba” -> “mamba”, “vision” -> “computer vision”)

Overall, I think that this paper should be revised carefully before it can be published on ICLR.

**Questions:**

I have some questions regarding the experiments, novelty, and the usefulness of this paper. Please see the above weaknesses for more details.

---

### Official Review · Reviewer_68YV · 2024-11-03

**Soundness:** 2
**Presentation:** 2
**Contribution:** 2
**Rating:** 5
**Confidence:** 4

**Summary:**

The manuscript explores the applicability of the Vision Mamba architecture in vision tasks, specifically comparing it with ConvNets. Through architecturally aligned comparisons, this work analyzes the effectiveness of different token mixers by replacing various token mixer modules, thereby investigating the necessity of SSM in vision tasks. Specifically, the study replaces the SS2D in VMamba with a DWConv(7x7) (referred to as GConvNet) and conducts strict comparative experiments using a controlled variable approach. Additionally, two hybrid strategies incorporating self-attention layers are proposed to investigate their impact on different models.

**Strengths:**

1. The study adopts an architecturally aligned comparison method that ensures fair and direct evaluation of token mixers by controlling for other variables. This approach enhances the reliability of conclusions drawn about the performance of VMamba and ConvNet architectures.
2.By leveraging state space models (SSM) as token mixers, VMamba provides a competitive alternative to traditional convolutional layers and self-attention mechanisms. The controlled comparisons indicate that VMamba’s token mixers outperform gated 7×7 depth-wise convolutions in tasks like image classification and object detection​.
3.VMamba can effectively integrate self-attention layers, enhancing its flexibility and applicability across different vision tasks. The inclusion of self-attention reduces overfitting in VMamba while improving the fitting capacity of GConvNet, making hybrid architectures a viable option for tasks requiring long-sequence modeling.

**Weaknesses:**

1.Although VMamba shows practical benefits, the manuscript lacks substantial new theoretical contributions. Many of the experimental findings resemble results seen in existing literature, with the primary value lying in comprehensive comparative studies rather than introducing fundamentally novel concepts.
2. Although the results indicate VMamba’s superior performance, there is limited explanation of the underlying reasons behind the token mixer’s effectiveness, especially compared to conventional ConvNets. A deeper exploration of the SSM token mixing mechanism and its interaction with self-attention layers would help clarify why VMamba performs well in specific tasks.
3. Despite its efficiency, the VMamba model may face scalability issues as task complexity increases. The manuscript does not address the adaptability of VMamba to lightweight models or discuss potential optimizations for reducing its computational footprint in large-scale real-world applications.

**Questions:**

1. The strict architecturally aligned comparison method ensures fairness in comparison by eliminating the interference of architectural hierarchy differences. This approach is reasonable, particularly after excluding other architectural variables, as it allows for a more direct assessment of the token mixer’s impact. However, since feature extraction requirements vary across tasks, whether this approach remains applicable in more complex scenarios is a question worth exploring in future research.
2. Although the paper provides extensive data on the comparison between Mamba models and ConvNets, much of the experimental results are similar to existing literature, lacking new theoretical or model innovations. The primary finding of the paper, that the Mamba model outperforms convolutional networks under specific experimental conditions, does not provide sufficient novelty. The manuscript reads more like a set of ablation studies rather than a breakthrough study.
3.  Please explain in detail the fundamental difference between Hybrid Models in Section 3.2 and ' VmambaIR: Visual State Space Model for Image Restoration '.
4. While the paper demonstrates Mamba’s performance on certain tasks through experiments, it lacks an in-depth theoretical analysis of why this model might outperform convolutional networks.

---

### Official Review · Reviewer_fadz · 2024-11-04

**Soundness:** 2
**Presentation:** 2
**Contribution:** 2
**Rating:** 3
**Confidence:** 5

**Summary:**

This paper explores the application of the Mamba architecture in vision tasks, particularly its performance in image classification and object detection. The authors demonstrates that Mamba's token mixers outperform traditional convolutional neural networks (ConvNets) on the ImageNet and COCO datasets, showcasing stronger expressivity and robustness. While incorporating a few self-attention layers in the top blocks can mitigate overfitting in VMambas on ImageNet. However, it is insufficient to bridge the performance gap between ConvNets and Vision Mambas.

The authors emphasizes the importance of architecturally aligned comparisons when introducing token mixers from natural language processing (NLP) into the vision domain. The findings suggest that the Mamba architecture has potential advantages in vision tasks, especially when combined with complex hierarchical models. Overall, the paper provides compelling evidence for the application of Mamba in the vision field and points to future research directions.

**Strengths:**

Stronger Expressivity: The architecture of VMamba allows for greater expressivity, which contributes to its ability to capture complex patterns in visual data more effectively than gated 7×7 depth-wise convolutions .

Robustness: VMambas exhibit superior robustness in natural image classification tasks, making them more reliable in various scenarios compared to their ConvNet counterparts .

**Weaknesses:**

1.To demonstrate the necessity and superiority of Mamba in vision, it is more suggested to design targeted improvements for Mamba instead of aligning architecture between ConvNet and Mamba. In vision tasks, different mechanisms are often suitable for different architectures. Exploring a more suitable architecte for Mamba to catch up or even surpass the current SOTA vision models is more meaningful and valuable.
2.Why not compare Mamba with Vision Transformers? This paper indicate that introducing Self-Attention layers can help VMamba mitigate overfitting. Under the same setting that introducing attention layers, VMamba surpasses ConvNets. This paper needs to compare Hybird Mamba and pure vision transformer model.
3.This paper may be more suitable as a technical report rather than a formal research publication. This paper only demonstrate the VMamba surpasses ConvNets under some architectures. The contribution is limited.
4. The improvement of Hybird VMamba comes from Transformer block. This does not indicate the necessity of introducing Mamba in vision tasks.

**Questions:**

There is redundancy presentation in Figure 7 which occupies a large amount of space. Combine bar charts of different models on the same dataset together will be more intuitive.

---

### Official Review · Reviewer_e9U6 · 2024-11-04

**Soundness:** 2
**Presentation:** 2
**Contribution:** 2
**Rating:** 5
**Confidence:** 3

**Summary:**

The paper focuses on comparing ConvNets and Vision Mambas in an architecturally aligned manner. It aims to answer the question that whether Mamba is necessary for vision tasks. The authors construct GConvNets and hybrid models, conduct experiments on ImageNet-1K and COCO datasets, and evaluate robustness. The results show: 1) VMamba's token mixers have advantages, 2) incorporating self-attention layers affects models differently, and 3) VMambas are generally more robust.

**Strengths:**

1. The findings are interesting. For example, the paper shows that VMamba's token mixers outperform gated 7×7 depth-wise convolutions in some cases, and uncovering the opposite effects of incorporating self-attention layers on VMambas and GConvNets (mitigating over-fitting in VMambas and enhancing fitting in GConvNets). The exploration of model robustness also provides useful insights.
2. The experiments are carried out on well-known datasets (ImageNet-1K and COCO) with clear protocols. By carefully controlling variables and constructing comparable models (like GConvNet), it provides a fair evaluation of the token mixers, which is a core contribution of Mamba. The author provides detailed experimental curve graphs.

**Weaknesses:**

1. Limited scope of comparison: while the focus of comparing token mixers is important, the paper could have explored more aspects of the architectures or performed comparisons with a wider range of existing models in the vision domain. In particular, mamba is a comparison to the transformer architecture. The author can also compare it to ViT.
2. As a structural analysis comparison paper, the authors could add more detailed comparisons and experimental verifications of the modules. In some cases, it would have been beneficial to conduct more ablation studies to further isolate the contributions of different components. For example, when discussing the impact of the MLP classifier in Mamba, a more detailed ablation analysis could have strengthened the conclusions.
3. Since Mamba has a great influence on the order of image sequences, the author should add verification on image segmentation.

**Questions:**

See weakness

---

### Meta-Review · Area_Chair_6AYw · 2024-12-20

**Metareview:**

(a) Scientific Claims and Findings

The paper investigates the applicability of the Vision Mamba architecture in vision tasks by comparing it with ConvNets through aligned experiments. The authors construct GConvNets and hybrid models, conducting experiments on ImageNet-1K and COCO datasets to evaluate robustness and performance. Reviewers e9U6, fadz, and 68YV note that the study finds VMamba's token mixers to have advantages over traditional convolutions, with self-attention layers affecting models differently and VMambas generally exhibiting more robustness.

(b) Strengths

Reviewer e9U6 appreciates the interesting findings, such as VMamba's token mixers outperforming gated convolutions and the exploration of model robustness. fadz highlights the stronger expressivity and robustness of VMambas, while 68YV commends the architecturally aligned comparison method for ensuring fair evaluation. P7hm notes the paper's clear motivation and professional writing, and ty4j finds the paper well-written and easy to follow.

(c) Weaknesses

The reviewers identify several weaknesses. e9U6 points out the limited scope of comparison and the need for more detailed module analysis. fadz suggests designing targeted improvements for Mamba and questions the necessity of the architecture alignment. 68YV criticizes the lack of new theoretical contributions and the limited explanation of VMamba's effectiveness. P7hm argues that the paper lacks technical novelty and should evaluate more models. ty4j finds the motivation unclear and the experimental organization lacking.

(d) Decision Reasons

The decision to reject the paper is based on the limited novelty and scope of the study, as highlighted by reviewers fadz and P7hm. While the paper provides interesting insights and a fair comparison method, the lack of new theoretical contributions and the limited exploration of VMamba's effectiveness reduce its impact. Additionally, the absence of a new model or technique and the unclear motivation and organization, as noted by ty4j, further weaken the case for acceptance. Despite the strengths in experimental validation, the weaknesses in innovation and scope lead to the decision to reject.

**Additional Comments On Reviewer Discussion:**

The authors have not provided a rebuttal to respond to the reviewer criticism, and all reviewers concur in their assessment that the paper is not ready for publication at ICLR.

---

### Decision · Program_Chairs · 2025-01-22

Reject